# Risk of Excess Maternal Folic Acid Supplementation in Offspring

**DOI:** 10.3390/nu16050755

**Published:** 2024-03-06

**Authors:** Xiguang Xu, Ziyu Zhang, Yu Lin, Hehuang Xie

**Affiliations:** 1Epigenomics and Computational Biology Lab, Fralin Life Sciences Institute, Virginia Tech, Blacksburg, VA 24061, USA; xiguang@vt.edu (X.X.); laurazhang@vt.edu (Z.Z.); yulin96@vt.edu (Y.L.); 2Department of Biomedical Sciences and Pathobiology, Virginia-Maryland College of Veterinary Medicine, Virginia Tech, Blacksburg, VA 24061, USA; 3Department of Human Development and Family Science, College of Liberal Arts and Human Sciences, Virginia Tech, Blacksburg, VA 24061, USA; 4Genetics, Bioinformatics and Computational Biology Program, Virginia Tech, Blacksburg, VA 24061, USA; 5Translational Biology, Medicine, and Health Program, Virginia Tech, Blacksburg, VA 24061, USA; 6School of Neuroscience, Virginia Tech, Blacksburg, VA 24061, USA

**Keywords:** folate, folic acid, maternal, neurodevelopment, gene expression, behavioral changes

## Abstract

Folate, also known as vitamin B9, facilitates the transfer of methyl groups among molecules, which is crucial for amino acid metabolism and nucleotide synthesis. Adequate maternal folate supplementation has been widely acknowledged for its pivotal role in promoting cell proliferation and preventing neural tube defects. However, in the post-fortification era, there has been a rising concern regarding an excess maternal intake of folic acid (FA), the synthetic form of folate. In this review, we focused on recent advancements in understanding the influence of excess maternal FA intake on offspring. For human studies, we summarized findings from clinical trials investigating the effects of periconceptional FA intake on neurodevelopment and molecular-level changes in offspring. For studies using mouse models, we compiled the impact of high maternal FA supplementation on gene expression and behavioral changes in offspring. In summary, excessive maternal folate intake could potentially have adverse effects on offspring. Overall, we highlighted concerns regarding elevated maternal folate status in the population, providing a comprehensive perspective on the potential adverse effects of excessive maternal FA supplementation on offspring.

## 1. Introduction

Folate is water-soluble vitamin B9 which includes the natural folate derived from food and the synthetic form. Folic acid (FA) is the synthetic form of folate and has been widely used in nutrient supplements and fortified food [1]. When consumed in our diet or supplements, folic acid is absorbed in the small intestine and then sequentially converted to dihydrofolate (DHF) and THF by dihydrofolate reductase (DHFR) in the cells [2]. Together with serine, THF can be reversibly converted to 5,10-methylene tetrahydrofolate (5,10-MTHF) and glycine by serine hydroxymethyltransferase (SHMT) [3,4]. With the aid of 5,10-methylenetetrahydrofolate reductase (MTHFR), 5,10-MTHF can be further catalyzed to 5-methyl THF (5-MTHF), a biologically active form of folate utilized in the synthesis of various molecules [5]. Food folate is the reduced form, and no reduction to dihydrofolate by DHFR is necessary [6]. More recently, another natural form (6S)-5-methyltetrahydrofolic acid (Metafolin^®^) has been used in place of the synthetic form folic acid, as it does not require either DHFR or MTHFR to form 5-MTHF [7]. 5-MTHF can be recycled back to THF via the re-methylation of homocysteine to form methionine and this process is crucial to maintaining an adequate supply of S-Adenosyl methionine (SAM), the key methyl donor in many biological methylation reactions (Figure 1). Additionally, 5-MTHF is required for nitric oxide synthesis via tetrahydrobiopterin and biogenic amine synthesis, producing some of the key neurotransmitters in the central nervous system [8,9].

Since folate is involved in the synthesis of nucleic acids and amino acids and is critical for cellular growth and differentiation [10], the demand for folate increases during pregnancy due to fetal/placental growth and uterus enlargement [11]. In addition, beyond the nervous system, the benefits of sufficient folate on reproductive and cardiovascular health have been demonstrated [12,13,14,15] and insufficient maternal folate further increases the risk of offspring overweight [16] and elevated blood pressure [17]. Additional adverse impacts of maternal folate deficiency have also been intensively reviewed in other research [18,19]. Due to food fortification and the relatively high use of multivitamin supplements, overall folate levels have significantly increased on a population-wide scale [20,21,22]. Recent studies raised concern regarding the adverse effect of maternal folate excess. For example, a ‘U shaped’ relationship was reported between maternal multivitamin supplementation frequency and the risk of Autism Spectrum Disorder (ASD); this association was further supported by the findings based on maternal plasma folate levels [23]. In this review, we center on recent advancements explored in both human studies and mouse models regarding the effects of high maternal folate intake and the molecular mechanisms influencing the health outcomes of offspring.

## 2. Human Studies on the Effect of Maternal Folic Acid Supplementation

Since the discovery of the association between folate deficiency and neural tube defects (NTDs), multiple clinical trials have confirmed the benefit of folate supplementation during pregnancy in reducing the incidence of NTDs [24,25,26]. As such, these studies have led to the recommendation of folic acid intake for pregnant women [27]. To increase compliance, the regulation of mandated fortification of grain products with folic acid was issued by the Food and Drug Administration in 1996 [28]. Current recommendations for folic acid supplementation for women are 400 μg/day before pregnancy, 600 μg/day during pregnancy, and 500 μg/day during lactation [29]. A significant reduction in NTD birth prevalence was achieved following the FA fortification of the US food supply [30], confirming the efficacy of food fortification in reducing NTD incidence in the population.

In the post-fortification era, serum and RBC folate levels have increased 2.5 times and 1.5 times, respectively, in the US population [21]. Periconceptional multivitamin or folic acid intake further increase folate levels in the pregnant women. In the Boston Birth Cohort, a wide range of plasma folate levels was shown in pregnant women, ranging from insufficient to excess levels [16]. The broad range of maternal folate intake enables dose–response studies to be conducted. Here, we have compiled recent human studies examining the effects of folate supplementation in the developmental outcomes of offspring (Table 1).

## 3. Influence of Periconceptional Folic Acid Supplementation in DNA Methylation in the Offspring

As a key player in methyl-donor metabolism, folic acid supplementation during pregnancy influences the DNA methylation profiles in offspring. Steegers-Theunissen et al. revealed the 4.5% higher methylation of the insulin-like growth factor 2 gene differentially methylation region (IGF2 DMR) in children born to mothers with periconceptional folic acid supplementation [51]. Moreover, the increased methylation level of IGF2 DMR was associated with decreased birth weight, indicating that FA-associated epigenetic changes in IGF2 in the child may affect intrauterine growth. Haggarty et al. also showed that folic acid use after 12 weeks of gestation was associated with higher methylation level in IGF2, and reduced methylation in both paternally expressed gene 3 (PEG3) and the long interspersed nuclear element 1 (LINE-1) [53]. Hoyo et al. assessed the association of maternal FA supplementation before and during pregnancy with aberrant DNA methylation at two DMRs regulating IGF2 and found decreased methylation levels at the IGF2/H19 DMR with increasing FA intake [52]. The inconsistent methylation changes at the IGF2 gene among different studies might be due to the different genomic regions studied. This also implies a complex relationship between maternal folic acid supplementation and the methylation alterations observed in offspring.

## 4. The Impact of Periconceptional Folic Acid Supplementation on the Neurodevelopment of Offspring

Adequate folate supplementation during pregnancy is considered positively associated with child neurodevelopment. Timmermans et al. revealed that periconceptional folic acid use was associated with higher placental and birth weight, and decreased risks of low birth weight and small for gestational age [33]. Eryilmaz et al. showed that the prenatal exposure of folic acid was associated with a cortical thickness increase in the bilateral, frontal, and temporal regions, as well as delayed age-associated cortical thinning in the temporal and parietal regions [34]. A randomized controlled trial (RCT) was performed to assess the effect of folic acid on fetal brain growth among pregnant women who smoke [35]. The results indicated that infants born to mothers who received high dose folic acid (4 mg/day) showed no difference in brain weight, but were 0.33 percentage points lower in brain/body weight ratio compared to those in the standard dose group (0.8 mg/day) [35].

In another RCT, known as the Folic Acid Supplementation in the Second and Third Trimesters trial, researchers found that continued FA supplementation during the second and third trimesters significantly increased the levels of maternal and cord RBC folate [31]. Compared to children born to mothers who only took 400 μg/day of FA during the first trimester, those whose mothers took 400 μg/day throughout the entire pregnancy achieved higher scores in cognition at 3 years old, and higher scores in word reasoning, emotional intelligence, and resilience at 6–7 years old [36,37]. Further follow-up investigations of this cohort showed that children from the latter group gained neurocognitive development benefits, as indicated by a higher score in two Processing Speed tests and more efficient semantic processing of language at 11 years old [38].

Multiple prospective cohort studies across different countries also support the beneficial effect of folate supplementation during pregnancy on child neurodevelopment. Julvez et al. showed that the maternal use of folic acid supplements was positively associated with verbal, motor, verbal-executive function, social competence, and inattention symptoms [39]. Veena et al. indicated a positive association between maternal plasma folate levels and children’s cognitive performance [40]. Roth et al. revealed that maternal use of folic acid in early pregnancy was associated with a reduced risk of severe language delay in children at the age of 3 years [41]. Chatzi et al. showed that children of mothers with reported doses of 5 mg/day or more folic acid had a 5-unit increase for receptive communication and a 3.5-unit increase in expressive communication [42]. Villamor et al. estimated that for every 600 ug/day increase in total folate intake during the first trimester of pregnancy, there was a 1.6-point increase in the cognition level measured by Peabody Picture Vocabulary Test III (PPVT-III) scores in the children at age 3 years [43].

Despite several studies supporting the benefit of periconceptional FA supplementation, some other prospective birth cohort studies have yielded inconsistent results. Wu et al. found no association between maternal plasma folate concentrations and child cognitive development at 18 months of age [44]. Boeke et al. found no association between maternal folate intake and cognitive outcomes in children aged 7 years [45]. Huang et al. revealed that maternal serum folate levels in late pregnancy were positively associated with children’s language development. Conversely, maternal serum folate levels in early pregnancy were inversely related to fine motor development in children at the age of 2 years [46]. Irvine et al. showed that maternal folate levels during the second trimester of pregnancy were linked to improved executive function development, but not associated with children’s intelligence, language, memory, or motor outcomes at 3–5 years of age [47]. A few retrospective cohort studies were performed, as well, to assess the benefit of maternal FA supplementation. Wehby and Murray showed that folic acid supplementation was associated with improved gross motor development. However, they showed a marginally significant negative association with performance in the personal–social domain [49]. Tamura et al. found no association between children’s cognitive development and maternal plasma or erythrocyte folate concentrations [48]. 

## 5. Dose-Dependent Association of Folic Acid Supplementation with the Incidence of ASD

In addition to general neurodevelopment in children, maternal folate intake also showed an association with the incidence of ASD. Surén et al. showed that folic acid intake from 4 weeks before to 8 weeks after the start of pregnancy was associated with a decreased risk of ASD in the prospective Norwegian Mother and Child Cohort [14]. More recently, data from the Boston Birth Cohort showed a “U-shaped” relationship between maternal multivitamin supplementation frequency and ASD risk [23]. Moderate self-reported supplementation during pregnancy was associated with a decreased risk of ASD, which is consistent with previous findings [14]. On the other hand, both low and high frequencies of multivitamin supplementation were linked to an elevated risk of ASD [23]. More precisely, an exceptionally high level of maternal plasma folate at birth demonstrated a 2.5 times increased risk of ASD incidence. Raghavan et al. further elucidated that a higher concentration of cord blood unmetabolized folic acid (UMFA), but not 5-methyl THF or total folate, was associated with the increased risk of ASD [50]. After stratifying by race, this UMFA-ASD association was restricted to Black children [50]. Although the precise cause remains elusive, speculation has arisen regarding genetic or metabolic variances in the processing of folic acid within cells [55]. More recently, a randomized clinical trial was performed to evaluate the impact of supplementation with either the synthetic form, folic acid, or the natural form, (6S)-5-methyltetrahydrofolic acid [(6S)-5-MTHF], on the human milk folate profile and it identified similar total milk folate but a significantly higher concentration of UMFA in those receiving folic acid versus the natural folate [32]. The above findings indicate a potential link between a high level of UMFA and the incidence of ASD. Further research is needed to investigate the underlying mechanisms.

## 6. Mouse Models to Study the Impact of High-Dose Maternal Folic Acid Supplementation

With a well-controlled experimental design, results from mouse models could provide insightful knowledge on the effect of high folate intake on offspring. When comparing results across different research groups (Table 2), various factors in the experimental schema should be considered. First, the timing and duration of FA supplementation vary among different studies and significantly impact the interpretation of study results. While most studies provided FA supplementation in adult female mice, others, such as Bahous et al. [56] and Luan et al. [57,58], began FA diets in female mice at weaning age (P21 and P30, respectively) and continued for specific durations (5 weeks and 30 days, respectively) before mating. Maternal FA supplementation typically comprises three phases: pre-mating, gestation, and lactation. The pre-mating period varies in duration, from as short as one week [59,60,61] to as long as six weeks [62], as in the study by Henzel et al., where female mice received FA diets for six weeks before mating and then returned to a control diet [62]. In contrast, some studies applied FA diets during all three phases to mimic the FA supplementation scenario in the human population [60,61,63].

Second, the concentrations of FA in both the control diets and the FA-supplemented diets vary among different studies. For instance, in an earlier study by Barua et al., the control diet contained 0.4 mg FA/kg [59,64,65], a level considered necessary for a normal healthy litter size [66]. In contrast, subsequent researchers reached a consensus that 2 mg FA/kg diet is the recommended level for rodents, and this criteria was applied to the most widely used with an AIN-93 diet [67]. As such, most recent studies used a control diet containing 2 mg/kg FA [56,57,58,60,61,63], with the exception of Henzel et al., who used a slightly different diet containing 2.3 mg/kg FA as the control [62]. For the FA-supplemented diet, Barua et al., who used 0.4 mg/kg FA for the control diet (CD), utilized 4 mg/kg FA, a tenfold higher level [59]. Meanwhile, Henzel et al. used 2.3 mg/kg FA as the control diet and used 40 mg/kg FA as the FA-supplemented diet, nearly 20 times higher [62]. Given that 2 mg/kg FA is considered to be the conventional level for rodents, a 10 times higher level of FA (20 mg/kg) is considered to be the high FA dosage [56,60,63]. Additionally, a few studies explored the influence of moderate FA supplementation in offspring and used either 2.5 times higher (5 mg/kg FA) [60,61] or 5 times higher (10 mg/kg FA) [57,58] levels of FA compared to the control diet. In this review, we excluded references that used 0.4 mg/kg FA as the control diet, as it falls outside the scope.

**Table 2 nutrients-16-00755-t002:** Summary of mouse studies on the influence of maternal high folate intake in the offspring. NA—not available.

Reference	FA Dosage	Duration	Gene Expression Changes	Behavioral Changes
Barua et al., 2015 [68]	2 mg/kg 20 mg/kg	1 week before gestation and throughout the pregnancy	Compared to the control, the P1 cerebellum of high FA group showed 1076 downregulated and 499 upregulated genes in male pups, and 4764 downregulated and 1511 upregulated genes in female pups, with 339 downregulated and 152 upregulated transcripts shared by both sexes.	NA
Barua et al., 2016 [69]	2 mg/kg 20 mg/kg	1 week prior to mating and throughout the entire period of gestation	RT-qPCR and Western blot analysis of P1 cerebral hemispheres in the high FA group showed sex-specific changes in transcription factors Nfix, Runx1, and Vgll2, DNA Methyltransferase Dnmt3b, the imprinted gene Dio3, H19, and Xist, and the candidate autism susceptible gene Auts2, Fmr1 at mRNA levels, and Gad1, Park2, and Hsp90 at protein levels.	NA
Bahous et al., 2017 [56]	2 mg/kg 20 mg/kg	5 weeks before mating, during pregnancy and lactation until P21	RT-qPCR analysis of cortex and hippocampus in P21 male offspring showed decreased Dnmt3a mRNA expression in the high FA group.	No differences were observed in ladder beam, open field, and Y-maze tests in HFA male P21 pups. In the novel object recognition test, HFA male pups exhibited short-term memory impairment, indicated by less time spent with the novel object compared to the familiar object.
Henzel et al., 2017 [62]	2.3 mg/kg 40 mg/kg	6 weeks prior to mating	RNA-seq of hippocampus at 14 weeks showed 12 differentially expressed genes, and the changes were diminished at 12 months old.	No differences were observed in open field, context fear conditioning, object place recognition, and rotarod tests in 14-week-old high FA offspring. However, in the Morris water maze test, high FA offspring showed impaired cognitive flexibility in reversal learning tasks.
Chu et al., 2019 [60]	Control: 2 mg/kg MFA: 2.5× FA HFA: 10× FA	1 week before mating, throughout pregnancy and lactation until P21	RNA-seq of cortex in the male offspring identified 176 DEGs (103 upregulated, 73 downregulated) in the MFA group compared to the control, and 96 DEGs (34 upregulated, 62 down regulated) in the HFA group compared to the control.	Compared to the control, MFA male offspring at 2 months old showed no differences in motor ability and spatial memory, whereas they displayed elevated anxiety-like behavior, impaired social preference, motor learning, and spatial learning ability. The HFA male offspring at 2 months old exhibited only mild behavioral abnormalities, without an effect on social behavior or anxiety-like behavior in the elevated plus maze.
Yang et al., 2019 [70]	Control: 2 mg/kg MFA: 5 mg/kg	Throughout pregnancy and lactation	RT-qPCR and Western blot analysis of hippocampus at 7 weeks showed increased expression of PCNA, DCX, BDNF, and GR in the MFA group.	Compared to the control, MFA male offspring at 6 weeks showed increased spatial learning and memory with fewer fear-related behaviors, represented by increased center area entries and duration in the open field test, and increased frequency of entering open arms in the elevated plus maze. No differences were observed in the forced swimming test or tail suspension test.
Yang et al., 2021 [61]	2 mg/kg MFA: 2.5× FA	1 week before mating, throughout pregnancy and lactation until P21	RNA-seq analysis of P21 cortex showed 115 DEGs (36 upregulated, and 79 downregulated) in the MFA female group compared to the control, with 39 DEGs (16 upregulated and 23 downregulated) overlapping with the DEGs identified in the MFA male group.	Compared to the control, the MFA female offspring at 2 months old showed decreased exploratory activity and increased anxiety-like behavior in the open field test, and impaired motor coordination in the rotarod test. No differences were found in the three-chamber social approach, social novelty test, or the elevated plus maze test.
Harlan De Crescenzo et al., 2021 [63]	0 mg/kg 2 mg/kg 20 mg/kg	2 weeks before mating, during pregnancy and lactation until P21	Western blot analysis of P0 brain and liver tissues showed no significant changes in Mthfr protein levels.	Offspring aged 4–10 weeks in the high FA group exhibited increased anxiety-like behavior in the elevated plus maze, reduced exploratory behavior and potential increase in anxiety in the open field test, and reduced marble burying in the marble burying test. No significant differences were noted in the novel object recognition and 3-chambered social approach tests.
Cosín-Tomás et al., 2020 [71]	2 mg/kg 10 mg/kg	1 month before mating, during pregnancy and lactation until P30	Western blot analysis of P30 cortex and liver in the high FA group showed a decreased expression of Mthfr protein levels in the liver, but not in the cortex.	The high FA offspring aged 3 weeks showed hyperactivity-like behaviors without notable differences in the anxiety levels in the open field test, and short-term memory impairment in the novel object recognition test. No differences were observed in grip strength or social interaction.
Luan et al., 2021 [57]	2 mg/kg 10 mg/kg	1 month before mating, during pregnancy until E17.5	Microarray analysis of the E17.5 placenta showed 186 DEGs in males and 274 DEGs in females, with only seven genes common between two sexes. RT-qPCR analysis confirmed changes in 29 genes associated with angiogenesis, receptor biology, and neurodevelopment. Western blot analysis did not show significant differences in MTHFR expression in the placenta.	NA
Luan et al., 2022 [58]	2 mg/kg 10 mg/kg	1 month before mating, during pregnancy and lactation until P30	Microarray analysis of the E17.5 cortex in the high FA group identified 274 DEGs (114 downregulated, 160 upregulated) in males, and 354 DEGs (177 downregulated, 177 upregulated) in females, with only 5 commonly changed genes in both sexes. Microarray analysis of the P30 cortex in the high FA group identified 599 DEGs (357 downregulated, 242 upregulated) in males and 419 DEGs (202 downregulated, 217 upregulated) in females, with only 15 genes shared in both sexes.	NA

Third, the approaches to supplementing FA also vary across different studies. Most studies supplemented the desired concentration of FA in the diet provided by the manufacturers. However, Chu, D. et al. and Yang, X. et al. from the same research group used the control diet containing 2 mg/kg FA for all mice and supplemented FA in the drinking water at the levels of 0 mg/L (control), 3.75 mg/L (MFA), or 22.5 mg/L (HFA) [60,61]. With an estimated daily consumption of 5 g diet and 4~8 mL water, the MFA and HFA groups received at least 2.5 times and 10 times more FA, respectively. In addition to FA supplementation, Bahous et al. also added the antibiotic succinylsulfathiazole at 1% to the diet, with the aim of inhibiting folate synthesis by intestinal flora in the mice [56]. In contrast, most studies did not mention the inclusion of antibiotics to the diet, probably due to the influence of intestinal flora on the folate level, which may be minimal in the context of high folate research.

Last but not least, sex is also a critical biological factor in folate study. Recent reports indicated sex difference in the influence of folate supplementation in the offspring. Barua et al. revealed that high gestational folic acid supplementation (10× FA) alters the gene expression in the P1 cortex and cerebellum in a sex-specific manner in mouse offspring [68,69]. Chu et al. and Yang et al. assessed the impact of excess FA supplementation (2.5× FA) before and during pregnancy and lactation in the P21 cortex of male and female offspring, respectively [60,61]. Luan et al. checked the sex-specific influence of moderate FA supplementation (5× FA) in E17.5 placenta, E17.5 cortex, and P30 cortex in both male and female offspring [57,58]. Sex-specific gene expression changes and behavioral abnormalities had been detected in the above studies. 

## 7. Gene Expression Changes Induced by Maternal High Folate Supplementation

RNA-seq, microarray, RT-qPCR, or Western blot have been utilized in several studies to assess RNA or protein expression changes at various developmental stages in offspring born to mothers with high folate supplementation. Barua et al. performed a microarray analysis of P1 cerebellum to evaluate the effect of high folate intake on the gene expression changes in newborn pups and found thousands of differentially expressed genes (DEGs) in both female and male offspring [68]. Specifically, they identified 1076 downregulated DEGs in the male pups and 4764 downregulated DEGs in the female pups, with 399 genes overlapping, and they identified 499 upregulated DEGs in the male and 1511 upregulated genes in the female, with 152 genes overlapping [68]. A functional analysis revealed a list of transcription factors and imprinted genes that were enriched for specific neural pathways, such as the dopaminergic and serotonergic pathways, GABA-glutamate pathway, and neurogenesis. And some of these expression changes varied in a gender-specific manner [68]. Further, Barua et al. performed RT-qPCR and Western blot analysis in P1 cerebral hemispheres of male and female offspring, and confirmed sex-specific alterations in expression in a wide range of genes, including transcription factors Nfix, Runx1, and Vgll2, DNA Methyltransferase Dnmt3b, the imprinted gene Dio3, H19 and Xist, and the candidate autism susceptible gene Auts2, Fmr1 at mRNA levels, and Gad1, Park2, and Hsp90 at protein levels [69]. The results indicate sex-specific gene expression changes induced by high gestational FA intake. 

Several studies have assessed the gene expression changes in the offspring at weaning age. Chu et al. initially performed RNA-seq in the P21 cortex of male offspring from three groups: control (2 mg/kg FA), MFA (5 mg/kg FA, 2.5× FA), and HFA (20 mg/kg FA, 10× FA). Surprisingly, the authors identified many more differentially expressed genes in the MFA group than the HFA group when compared to the control group (176 DEGs in the comparison between MFA and control, 96 DEGs in the comparison between HFA and control) [60]. As such, the authors focused on the influence of a moderate dose of FA and identified 24 genes differentially expressed exclusively in the MFA group. These genes were enriched for the Fos-related regulatory network and were critical for behavior and synaptic transmission [60]. Further, Yang et al. from the same research group performed RNA-seq to assess the influence of moderate periconceptional FA supplementation in the P21 cortex of female offspring and identified 115 DEGs, with 39 genes overlapping with DEGs identified in male offspring [61], indicating sex-specific influence by moderate FA supplementation. Interestingly, functional annotation showed that the DEGs induced by MFA in both male and female offspring were enriched for several shared pathways, including cocaine addiction, amphetamine addiction, and alcoholism [60,61], indicating a potential influence of maternal folate supplementation in the reward system.

In another study, Luan et al. performed a microarray analysis to assess the influence of moderate FA supplementation (10 mg/kg FA, 5× FA) in the E17.5 placenta, E17.5 cortex, and P30 cortex of both male and female offspring and found sex-specific transcriptional changes in the placentas, embryonic, and early postnatal brains [57,58]. Specifically, the authors identified 186 DEGs in the E17.5 placenta of male offspring and 274 DEGs in the E17.5 placenta of female offspring, with only 7 genes overlapped between two sexes [57]. Likewise, the authors identified 274 DEGs in the E17.5 cortex of male offspring and 354 DEGs in the E17.5 cortex of female offspring, with only 5 commonly changed genes between the two sexes [58]. Furthermore, the authors identified 599 DEGs in the P30 cortex of male offspring and 419 DEGs in the P30 cortex of female offspring, with only 15 genes shared between the two sexes [58]. Such a low rate of common DEGs between male and female offspring indicates a sex-specific influence of maternal FA supplementation in offspring. 

Additionally, Henzel et al. checked the gene expression changes in the adult offspring. Henzel et al. performed the RNA-seq of the hippocampus from 14-week or 12-month offspring of control (2.3 mg/kg FA) and FA-supplemented (40 mg/kg FA, 20× FA) groups, only to identify 12 differentially expressed genes in the 14-week offspring, and no differentially expressed genes identified in the 12-month offspring [62]. This is probably due to the FA diets being restricted to a period prior to mating and that the gene expression changes might diminish with age. Collectively, the above findings from different research groups indicate that high maternal folate intake induces significant gene expression changes in a sex-specific manner. These aberrant gene expression changes were predominant in the newborn and young offspring, and may influence early postnatal brain development, resulting in long-lasting behavioral changes in adult offspring.

## 8. Developmental and Behavioral Abnormalities Induced by Maternal High Folate Supplementation

It is well known that maternal folate deficiency can lead to neural tube defect, including conditions like spina bifida and anencephaly [19]. Intriguingly, excess maternal folate supplementation may also induce neurodevelopmental changes in offspring. Offspring from dams receiving 10 times the normal folate supplementation showed reduced embryo and placental weights at E17.5, along with decreased thickness in the dentate gyrus (DG) and a smaller hippocampal area in P21 offspring [56]. Harlan et al. noted a significant thinning of the cerebral cortex at P6, particularly at the dorsal aspects [63]. Conversely, offspring from dams receiving 5 times the normal folate supplementation displayed no changes in embryonic or postnatal growth [57,71]. For the offspring from dams receiving 2.5 times the normal folate supplementation, Chu D. et al. conducted a 5-month postnatal weight assessment. They found no difference in weaning weight at P21 but observed increased body weight gain in male mice at P60 and P150. This effect was not observed in female offspring, suggesting a sex-specific impact of early-life FA supplementation on physical growth [60,61].

In addition to the developmental changes, multiple studies have indicated that excess maternal high folate supplementation can also induce behavioral changes in offspring. Bahous, R. H. et al. observed that three-week-old pups born to mothers who received 10 times the normal folate supplementation exhibited short-term memory impairment [56]. In a study by Cosín-Tomás, M. et al., maternal supplementation with 5 times the normal amount of folate led to hyperactivity-like behavior and short-term memory impairment in the 3-week-old offspring of both sexes [71]. Bahous, R. H. et al. and Cosín-Tomás, M. et al. performed behavioral tests in young mice at weaning age [56,71], while in the following mice studies, behavioral tests were performed in adult offspring. The detailed behavioral changes were summarized in Table 2. Briefly, Harlan A. et al. conducted behavioral assessments in young mice (4–6 weeks old) and confirmed behavioral deviations in offspring born to mothers with excess folate intake. These deviations included increased anxiety and repetitive behavior, but no significant changes in short-term memory or social interactions [63]. Henzel KS et al. restricted folate diets to a period before mating and identified reduced reversal learning abilities in adult offspring (14 weeks old) of the excess folate group (20 times the normal intake), which is indicative of behavior associated with ASD [62]. Chu D. et al. evaluated the influence of maternal FA supplementation on the adult offspring using a battery of behavior tests and found that moderate folate intake (2.5× FA) but not HFA (10× FA) led to elevated anxiety-like behavior and impaired social activity but had no difference in performance in social novelty, impaired motor learning, and spatial learning in the male adult offspring [60]. HFA (10× FA) only showed more exploratory behavior compared to the controls [60]. Further, Yang X. et al., from the same research group, examined the influence of moderate folate supplementation in female offspring and found increased anxiety-like behavior, reduced exploratory activity, motor learning, and spatial learning, with no changes in sociability. Meanwhile, Yang Y. et al.’s maternal FA supplementation (2.5× FA) throughout gestation and lactation enhanced adult male offspring learning and memory with less fear-related behavior [70]. Ryan et al. applied a methyl donor-rich diet (MD, containing a supplement of 15 mg/kg FA) or control diet (CD, containing 4 mg/kg folate) to the male mice 6 weeks prior to mating, and found that the F1 offspring of MD fathers showed impaired hippocampus-dependent learning and memory, such as reduced target quadrant occupancy and target crossings in the Morris water maze, and impaired contextual fear conditioning [72]. These findings collectively highlight the long-term influence of excess maternal folate intake on behavioral alterations in offspring.

## 9. Metabolic and Reproduction Disorders Induced by Maternal High Folate Supplementation

In addition to its influence on the nervous system, maternal high folate supplementation could also induce metabolic and reproduction disorders in offspring. Huang et al. applied a control diet or high FA (20× FA) diet 2 weeks before and throughout pregnancy to evaluate the effect of maternal FA supplement on glucose metabolism in offspring. The authors found that, compared with the controls, male offspring derived from high FA dams had a higher risk of developing obesity, glucose intolerance, and insulin resistance under high fat diet feeding conditions [73]. Further, Kintaka et al. used the same concentration of high FA (20× FA) during pregnancy to determine its effect on the glucose tolerance in offspring. The authors found that excess maternal FA supplementation results in lower insulin synthesis and aberrant gene expression changes related to hepatic fat metabolism in offspring [74]. Mussai et al. evaluated the influence of maternal folic acid supplementation before and during pregnancy using a Western diet-induced diabetes model. The authors found that 5× FA supplementation led to a greater beta cell mass and density in the pancreas of male but not female offspring, accompanied with sex-specific hepatic gene expression changes in the E18.5 fetal offspring [75]. Sun et al. evaluated the effect of FA supplementation on glucose metabolism disorders in male offspring induced by lipopolysaccharide exposure during pregnancy. The authors found that 2.5× FA but not 20× FA supplementation during pregnancy and lactation improved glucose metabolism in lipopolysaccharide-exposed male offspring compared to the controls [76]. Collectively, the above results demonstrate a significant influence of excess maternal FA supplementation in glucose metabolism in offspring.

Rahimi et al. evaluated the influence of moderate and high maternal FA supplementation in assisted reproductive technologies (ART) by applying 4× FA or 10× FA or control (2 mg/kg FA) diets 6 weeks prior to ART and throughout gestation. The authors found that moderate FA (4× FA) supplementation in ART was associated with beneficial effects, including a decreased proportion of embryonic developmental delay, and reduced DNA methylation variance at imprinted control regions (ICRs) in the placenta and embryo tissues, while a high dose of FA (10× FA) supplementation exacerbated the negative effects of ART, as it further increased the methylation variance at ICRs and further decreased the mean methylation level at certain ICRs [77]. Ly et al. assessed the influence of FA supplementation in male germ cells and found that F1 males with a lifetime exposure to 20× FA exhibited lower sperm counts, increased post-implantation losses among F2 E18.5 litters, and had higher postnatal–preweaning pup death [77]. Further, Chan et al. from the same research group compared the DNA methylation profiles in germ cells of male offspring across three generations, with only F1 male offspring exposed to lifetime FA supplementation. The authors found that F1 germ cells showed the greatest number of differentially methylated CpG sites, and the altered methylation of specific sites in F1 germ cells was not present in later generations, indicating no transgenerational inheritance effects in maternal folate supplementation [78]. Altogether, the above findings indicate widespread effects of maternal high folate supplementation on offspring.

## 10. Discussion and Future Directions

In the post-fortification era, an elevated maternal folate status has increasingly become prevalent among the population. In this review, we assessed the influence of excess maternal folate intake on offspring, examining reports from both human and mouse studies. We specifically focused on its effects on the central nervous system. In human studies, a wide range of folate levels has been observed, and the correlation between folate levels and neurodevelopment has been evaluated. Both beneficial and detrimental effects have been identified. Given the diverse genetic background of the population, additional consideration should be taken regarding folate supplementation during pregnancy and lactation. In mouse studies, the aberrant influence of high maternal folic acid supplementation in gene expression and behavioral changes in offspring was observed in a sex-specific manner. This raises concerns that excessive folate intake during pregnancy could pose risks.

Despite significant progress having been made in our understanding of the adverse effects of excess maternal folate, there are significant gaps in our knowledge regarding the following areas: (1) How do we translate findings from mouse models to human studies and vice versa? Mouse models offer a consistent genetic background, controlled experimental environment, and accessibility to tissues. Due to many species-specific features related to gestation and neurodevelopment, comparative genomics studies with cross-species validation are highly desired to identify conserved molecular pathways relevant to maternal folate intake. (2) Alternative forms of maternal folate supplementation. Recent studies have shown promising supplementation with alternate forms of folate aiming at maintaining the benefit of NTD prevention while mitigating the potential harm of excess circulating UMFA. This also highlights the need to carefully examine the correlation of phenotypes with various forms of folate metabolic intermediates. (3) Several mouse models have demonstrated that high maternal folate supplementation can influence offspring in a sex-specific manner, affecting both gene expression changes and behavioral abnormalities. Given the strong association between maternal folate intake and ASD, along with the significant sex bias in ASD incidence rates, future research on the sex-linked impact of maternal folate intake is warranted. With the recent advancements in tools in systems biology, we anticipate a better understanding of the molecular mechanisms underlying maternal folate-associated disorders.

## Figures and Tables

**Figure 1 nutrients-16-00755-f001:**
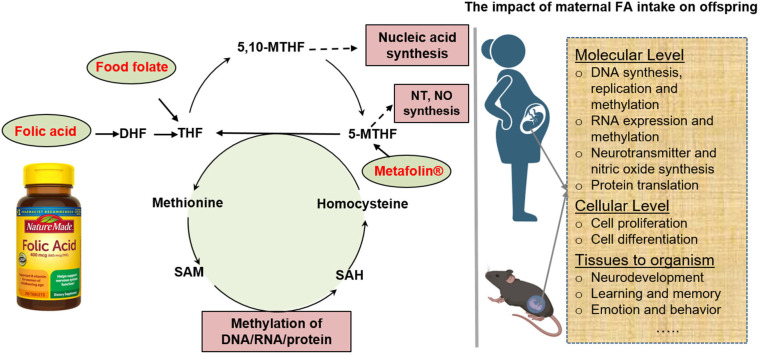
A simplified illustration of FA metabolism and the impact of maternal intake on offspring. Folate from food or folic acid (FA) from fortified food/supplements are converted by dihydrofolate reductase to form dihydrofolate (DHF) and then tetrahydrofolate (THF). THF is then converted to 5,10-methylenetetrahydrofolate (5,10-MTHF), which is required for nucleic acid synthesis. 5,10-MTHF is further converted by methylenetetrahydrofolate reductase to form 5-methyltetrahydrofolate (5-MTHF), which is necessary for neurotransmitter (NT) and nitric oxide (NO) synthesis. 5-MTHF is converted by methionine synthase back to THF. The methyl group is transferred from 5-MTHF to homocysteine to form methionine. Methionine can be converted to S-adenosylmethionine (SAM) and acts as the methyl donor in the methylation reactions. SAH, S-adenosylhomocysteine. Excess maternal FA intake induces changes at molecular, cellular, tissue, and organism levels in offspring.

**Table 1 nutrients-16-00755-t001:** Human studies on the influence of folic acid supplementation during pregnancy in the offspring. NA—not available. Background colors categorize references into groups.

Reference	Country	Sample Size	Research Type	Assessment of Folic Acid Use (Dosage and Stage)	Measurements	Folate Level (nmol/L)	Key Findings
McNulty et al., 2013 [31]	United Kingdom	59 women in the folic acid group, 60 women in the placebo group	Randomized controlled trial (RCT)	All women took 400 μg/d of FA in the first trimester, and received 400 μg/d folic acid or placebo during the second and third trimesters.	Serum and red blood cell folate, serum vitamin B-12, and plasma homocysteine were analyzed.	Serum folate: 45.7 ± 21.3 in placebo group and 47.0 ± 21.0 in treatment group. Red blood cell folate: 1106 ± 746 in placebo group and 1203 ± 639 in treatment group.	Continued FA supplementation during the second and third trimesters significantly increased the levels of maternal and cord red blood folate, and decrease the level of plasma homocysteine.
Cochrane et al., 2023 [32]	Canada	60 mother–child pairs	Randomized control trials (RCT)	Pregnant women were enrolled at 8–21 weeks of gestation and randomized to 0.6 mg/day folic acid or an equimolar dose (0.625 mg) of (6S)-5-methyltetrahydrofolic acid [(6S)-5-MTHF].	Folate and cord blood unmetabolized folic acid (UMFA) in human milk collected 1 week postpartum were quantified via LC–MS/MS.	Total folate in human milk: 47 ± 20 in (6S)-5-MTHF group and 61 ± 28 in folic acid group. UMFA in human milk: 0.6 in (6S)-5-MTHF group and 12 in folic acid group.	Compared to natural folate, folic acid supplementation showed similar levels of total folate, but significantly higher levels of UMFA in the milk.
Timmermans et al., 2009 [33]	Netherlands	6353 pregnancies	Prospective birth cohort	Self-reported questionnaires on folic acid use.	Fetal growth measured in mid- and late pregnancy by ultrasound; birth weight, small for gestational age (SGA), and preterm birth were recorded at birth.	NA	Periconceptional folic acid use was associated with increased fetal growth resulting in higher placental and birth weight, and decreased risks of low birth weight and small for gestational age (SGA).
Eryilmaz et al., 2018 [34]	United States	292 youths 8–18 years of age	Retrospective clinical cohort	None, partial, or full prenatal folic acid fortification exposure.	Cortical thickness was measured by brain MRI scans.	NA	Prenatal exposure of folic acid was associated with cortical thickness increase in the bilateral, frontal, and temporal regions, as well as delayed age-associated cortical thinning in the temporal and parietal regions.
Yusuf et al., 2019 [35]	United States	345 smoking pregnant women	Randomized control trials (RCT)	Participants were randomly assigned to receive either 0.8 mg folic acid/d or 4 mg folic acid/d.	Fetal growth was assessed by intrauterine ultrasound.	NA	Infants of mother who received high dose folic acid showed no difference in brain weight, but were 0.33 percentage points lower in brain/body weight ratio compared to standard dose group.
Henry et al., 2018 [36]	United Kingdom	22 mother–child pairs in the folic acid group and 17 mother–child pairs in the placebo group	Randomized controlled trial (RCT)	All women took 400 μg/d of FA in the first trimester, and received 400 μg/d folic acid or placebo during the second and third trimesters.	Emotional intelligence and resilience were assessed in children at 6–7 years.	NA	Children of folic-acid-treated mothers had higher scores in emotional intelligence and resilience at 6–7 years old.
McNulty et al., 2019 [37]	United Kingdom	37 mother–child pairs in the folic acid group and 33 mother–child pairs in the placebo group	Randomized controlled trial (RCT)	All women took 400 μg/d of FA in the first trimester, and received 400 μg/d folic acid or placebo during the second and third trimesters	Cognitive performance was evaluated in children at 3 and 7 years.	NA	Children of folic-acid-treated mothers scored higher in cognition at 3 years old and had higher scores in word reasoning at 7 years old.
Caffrey et al., 2021 [38]	United Kingdom	68 mother–child pairs (37 in FA group, 31 in placebo group)	Randomized controlled trial (RCT)	All women took 400 μg/d of FA in the first trimester, and received 400 μg/d folic acid or placebo during the second and third trimesters.	Cognitive performance was assessed by the Wechsler Intelligence Scale in children at 11 years old.	NA	Children of folic-acid-treated mothers scored higher in two Processing Speed tests, and showed more efficient semantic processing of language at 11 years old.
Julvez et al., 2009 [39]	Spain	420 mother–child pairs	Prospective birth cohort	Interviewer-administered questionnaires at the end of the first trimester of pregnancy.	Psychological outcomes were assessed in children at age 4 years.	NA	Maternal use of folic acid supplements was positively associated with verbal, motor, verbal-executive function, social competence, and inattention symptoms.
Veena et al., 2010 [40]	India	536 mother–child pairs	Prospective Mysore Parthenon birth cohort	Folate levels were measured from maternal plasma samples collected at 30 weeks of gestation.	Cognitive function was measured in children at 9–10 years.	NA	It showed a positive association between maternal plasma folate levels and children’s cognitive performance.
Roth et al., 2011 [41]	Norway	38,954 mother–child pairs	Prospective mother–child cohort	Self-report 3-year follow-up questionnaires.	Children’s language competency at age 3 years measured by maternal report.	NA	Maternal use of folic acid in early pregnancy was associated with a reduced risk of severe language delay in children at the age of 3 years.
Chatzi et al., 2012 [42]	Greece	553 mother–child pairs	Prospective mother–child cohort	Interviewer-administered questionnaires at 14–18 weeks of gestation.	Neurodevelopment was assessed in children at 18 months.	NA	Children of mothers with reported doses of 5 mg/d or more folic acid had a 5-unit increase in receptive communication and a 3.5-unit increase in expressive communication.
Villamor et al., 2012 [43]	United States	1210 mother–child pairs	Prospective pre-birth cohort	Questionnaires on the use of food frequency during the first and second trimesters of pregnancy.	The cognition and visual-motor skills were assessed in children at age 3 years.	NA	For every 600 ug/day increase in total folate intake during the first trimester of pregnancy, there was a 1.6-point increase in the PPVT-III scores in the children at age 3 years.
Wu et al., 2012 [44]	Canada	154 mother–child pairs	Prospective birth cohort	Folate levels were measured from maternal plasma samples collected at 16 and 36 weeks of gestation.	Neurodevelopment was assessed in children at 18 months of age.	NA	No association was found between maternal plasma folate concentrations and child cognitive development.
Boeke et al., 2013 [45]	United States	895 mother–child pairs	Prospective pre-birth cohort	Semiquantitative food frequency questionnaire (FFQ) at each of the first- and second-trimester study visits.	Visual memory and verbal and non-verbal intelligence were assessed in children at the age of 7 years.	NA	No associations were found between maternal folate intake and the cognitive outcomes in children aged 7 years.
Huang et al., 2020 [46]	China	180 mother–child pairs	Prospective birth cohort	Serum folate concentrations were measured in blood samples collected from pregnant women at early, middle and late stages of pregnancy.	Gross motor skills, fine motor skills, language, adaptive behavior, and social behavior were assessed in children at the age of 2 years.	NA	Maternal serum folate in late pregnancy was positively associated with children’s language development while maternal serum folate in early pregnancy was inversely related to fine motor development in the children at the age of 2 years.
Irvine et al., 2023 [47]	Canada	309 mother–child pairs	Prospective birth cohort	Maternal RBC folate status assessed during the second trimester of pregnancy.	Neurodevelopment was assessed in children at 3–5 years old.	NA	Maternal folate status during the second trimester of pregnancy was associated with improved executive function development, but not associated with children’s intelligence, language, memory, or motor outcomes at 3–5 years of age.
Tamura et al., 2005 [48]	United States	335 mother–child pairs	Retrospective birth cohort	Both plasma and whole-blood folate concentrations were measured from maternal blood samples collected at 19, 26, and 37 weeks of gestation.	Six tests were performed in children at a mean of 5.3 years to assess their neurodevelopment.	NA	No association was found between maternal plasma and erythrocyte folate concentrations and children’s cognitive development.
Wehby and Murray, 2008 [49]	United States	6774 mother–child pairs	Retrospective birth cohort	The 1988 National Maternal Infant Health Survey (NMIHS) and its 1991 Follow-up Survey data.	16 Denver developmental screening items were measured in children at about 3 years of age.	NA	Folic acid use was associated with improved gross-motor development, but had marginally significant poorer performance for the personal–social domain.
Surén et al., 2013 [14]	Norway	85,176 children	Prospective birth cohort	The information of mothers’ supplement intake before conception and in early pregnancy was obtained through questionnaire report at week 18 of gestation.	Cases of ASD were diagnosed and confirmed by the health specialists.	NA	Prenatal folic acid supplementation around the time of conception was associated with a lower risk of ASD incidence.
Raghavan et al., 2018 [23]	United States	1257 mother–child pairs	Prospective birth cohort	A standard questionnaire was used to collect maternal data including supplement intake. Maternal plasma folate was measured from maternal blood samples collected 24–72 h post-delivery.	Children were diagnosed with ASD by the health specialists.	NA	It showed a “U shaped” relationship between maternal multivitamin supplementation frequency and ASD risk: moderate self-reported supplementation during pregnancy was associated with decreased risk of ASD, while low and high supplementation was associated with increased risk of ASD.
Raghavan et al., 2020 [50]	United States	567 mother–child pairs	Prospective birth cohort	A standard questionnaire was used to collect maternal data. Plasma and RBC folate levels were measured from umbilical cord blood samples collected at the time of delivery.	Children were diagnosed with ASD by the health specialists.	NA	Higher concentration of cord blood unmetabolized folic acid (UMFA) was associated with increased risk of ASD.
Steegers-Theunissen et al., 2009 [51]	Netherlands	120 mother–child pairs (86 mothers had used and 34 had not used folic acid periconceptionally)	Cross-sectional study	Questionnaire data via the mother on periconceptional folic acid use.	DNA methylation of IGF2 and folate levels in serum and red blood cells were measured using a mass spectrometry-based method in children between 12 and 18 months of age.	Serum folate in mothers: 15.3 in no FA group and 17.8 in yes FA group; Serum folate in children: 31.5 in no FA group and 32.1 in yes FA group; RBC folate in mothers: 687 in no FA group and 720 in yes FA group; RBC folate in children: 973 in no FA group and 1064 in yes FA group.	Children of mothers with periconceptional folic acid use had a 4.5% higher methylation of the IGF2 DMR and decreased birth weight.
Hoyo et al., 2011 [52]	United States	438 pregnancies	Prospective cohort study	Preconception and prenatal FA supplementation was assessed from the self-administered questionnaire.	The methylation levels of two IGF2 DMRs measured via pro-sequencing in umbilical cord blood leukocytes.	NA	The methylation level of the IGF2/H19 imprinted region decreased with increasing FA intake before and during pregnancy.
Haggarty et al., 2013 [53]	United Kingdom	913 mother–child pairs	Prospective cohort study	Folate levels were measured in maternal blood samples collected at 19 weeks of gestation and from cord blood samples.	DNA methylation level of 3 maternally methylated imprinted genes and 1 retrotransposon was measured in cord blood samples.	Maternal RBC folate: 456 Cord RBC folate: 657.	Folic acid supplement after 12 weeks of gestation was associated with a higher level of methylation in IGF2 and reduced methylation in both PEG3 and LINE-1.
Ondicova M et al., 2022 [54]	United Kingdom	86 cord blood DNA samples	Randomized controlled trial (RCT)	All women took 400 μg/d of FA in the first trimester, and received 400 μg/d folic acid or placebo during the second and third trimesters.	Methylation profiles of cord blood were measured by the EPIC array and validated using pyrosequencing.	NA	FA supplementation resulted in significant methylation changes at specific classes of neurodevelopmental genes in the cord blood.

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
