# Peer review of "Risk of Excess Maternal Folic Acid Supplementation in Offspring"

_nutrients, 2024, doi:10.3390/nu16050755_

Round 1

Reviewer 1 Report

Comments and Suggestions for Authors

When discussing folate metabolism the authors imply that folic acid and folate are interchangeable terms, e.g. in Figure 1, whereas they also acknowledge that folic acid is a synthetic form. They correctly say that folic acid enters folate metabolism via DHF and THF, and come to 5-methyl THF via 5.10.methylene THF, but fail to mention that this is not natural folate metabolism. Natural folate is 5-methyTHF and so by-passes MTHFR (that represents a block in metabolism in around 30% of the population) and methylates homocysteine immediately before entering the two parts of folate metabolism. This is an important consideration when discussing the negative effects of folic acid.

5methylTHF also feeds into neurotransmitter and nitric oxide synthesis via tetrahydrobiopterin, giving multiple hits as a consequence of folic acid supplementation and an MTHFR mutation for example

Good set of references included in the study showing range of effects of folic acid supplementation. Would the authors care to comment on whether studies are biased by their authors targeting specific effects rather than analysing global effects of folic acid? Also, the authors have focused on mouse studies, is there a reason to exclude the studies on rats, for example on the HTx rat model of congenital hydrocephalus that shows negative effects of folic acid on susceptible fetuses?

The positive benefits of folic acid supplementation are well described as are the contradictory studies in humans.

It would useful to have a table of genes and the affects attributed to maternal supplementation to compliment the detailed description in the text.

The future perspectives are good.

A discussion may be suggested in the quality of supplements used in the various studies. As supplements claiming to be food supplements or natural products are not regulated, as well as the stability of folates as chemicals, this may explain some of the contradictory data observed in the literature but this cannot be assessed as source of supplements is rarely given.

Overall, this review of the literature highlights both positive and negative effects of maternal folic acid supplementation. Recent evidence suggests that folate supplementation of males may also have benefit so that it is perhaps not surprising to find reports of benefits, not only on neural tube defects, but also congenital heart defects, cleft palate, cardiovascular issues and certain cancers.

Author Response

We appreciate the reviewers for all the valuable and constructive suggestions.

Reviewer 1

  1. When discussing folate metabolism the authors imply that folic acid and folate are interchangeable terms, e.g. in Figure 1, whereas they also acknowledge that folic acid is a synthetic form. They correctly say that folic acid enters folate metabolism via DHF and THF, and come to 5-methyl THF via 5.10.methylene THF, but fail to mention that this is not natural folate metabolism. Natural folate is 5-methyTHF and so by-passes MTHFR (that represents a block in metabolism in around 30% of the population) and methylates homocysteine immediately before entering the two parts of folate metabolism. This is an important consideration when discussing the negative effects of folic acid.

We really appreciate the reviewer for pointing out the difference in metabolism for different forms of folate. We added the metabolism of food folate and another natural form of folate (6S)-5-methyltetrahydrofolic acid (Metafolin®) in the introduction section.

We modified Figure 1 accordingly.

  1. 5methylTHF also feeds into neurotransmitter and nitric oxide synthesis via tetrahydrobiopterin, giving multiple hits as a consequence of folic acid supplementation and an MTHFR mutation for example

Thank you for add this point. We added the above functions of 5-methyl THF in the introduction section.

  1. Good set of references included in the study showing range of effects of folic acid supplementation. Would the authors care to comment on whether studies are biased by their authors targeting specific effects rather than analysing global effects of folic acid? Also, the authors have focused on mouse studies, is there a reason to exclude the studies on rats, for example on the HTx rat model of congenital hydrocephalus that shows negative effects of folic acid on susceptible fetuses?

We thank the reviewer for affirming the inclusion of a wide range of references. We tried not to comment the bias of specific studies, but rather compile studies from different aspects to review the diverse influence of excess maternal folate intake in the offspring, especially focusing on the influence in the central nervous system.

We thank the reviewer for point out the rat studies. Both rat and mouse belong to rodents with a lot of similarities. To avoid repeating, we focused on human and mouse studies in this review.

  1. The positive benefits of folic acid supplementation are well described as are the contradictory studies in humans.

We thank the reviewer for endorsing this point.

  1. It would useful to have a table of genes and the affects attributed to maternal supplementation to compliment the detailed description in the text.

We thank the reviewer for raising this point. In table 2 of mouse studies, we listed the genes with expression changes induced by maternal folate supplementation.

  1. The future perspectives are good.

We thank the reviewer for affirming this point.

  1. A discussion may be suggested in the quality of supplements used in the various studies. As supplements claiming to be food supplements or natural products are not regulated, as well as the stability of folates as chemicals, this may explain some of the contradictory data observed in the literature but this cannot be assessed as source of supplements is rarely given.

We thank the reviewer for raising this point. We added a discussion section.

Overall, this review of the literature highlights both positive and negative effects of maternal folic acid supplementation. Recent evidence suggests that folate supplementation of males may also have benefit so that it is perhaps not surprising to find reports of benefits, not only on neural tube defects, but also congenital heart defects, cleft palate, cardiovascular issues and certain cancers.

We thank the reviewer for raising the point that folate supplementation has more beneficial effects in both male and female. This review focused on the influence of maternal folate intake in the offspring, with a special focus on the influence in the central nervous system. As such, the influence in other systems is out of the scope of this review.

Reviewer 2 Report

Comments and Suggestions for Authors

I recommend that the abstract should be rewritten with conclusions drawn by the authors based on published data reviewed and not simply state that they have reviewed.

Author Response

Reviewer 2

I recommend that the abstract should be rewritten with conclusions drawn by the authors based on published data reviewed and not simply state that they have reviewed.

We thank the reviewer for raising the point. We have modified the abstract to include a conclusion sentence.

Reviewer 3 Report

Comments and Suggestions for Authors

Thank you for giving me the opportunity for review the manuscript entitled “Risk of Excess Maternal Folic Acid Supplementation in Offspring”.

The manuscript is interesting and in scope of the Journal however it requires some clarifications.

The topic is of interest, as pregnant women have special dietary requirements that need to be met to prevent damage to the growing fetus as well as to prevent the development of certain diseases later in live.

Please find the specific comments below:

1.       Most importantly, was this review registered in PROSPERO?  And was a quality instrument used to assess the quality of the included studies?

2.       The authors should add a table to the manuscript that contains the folate concentration values found in the study material. This will provide a clear and concise presentation of the data, making it easier for readers to understand and interpret the results

Author Response

Thank you for giving me the opportunity for review the manuscript entitled “Risk of Excess Maternal Folic Acid Supplementation in Offspring”.

The manuscript is interesting and in scope of the Journal however it requires some clarifications.

The topic is of interest, as pregnant women have special dietary requirements that need to be met to prevent damage to the growing fetus as well as to prevent the development of certain diseases later in live.

We thank the reviewer for affirming the significance of the topic.

Please find the specific comments below:

  1. Most importantly, was this review registered in PROSPERO?  And was a quality instrument used to assess the quality of the included studies?

We thank the reviewer for raising the point. As this is not a meta-analysis or systematic review, we included the peer-reviewed original studies in human and mouse models. We didn’t apply specific quality instrument to assess the quality of the included studies.

  1. The authors should add a table to the manuscript that contains the folate concentration values found in the study material. This will provide a clear and concise presentation of the data, making it easier for readers to understand and interpret the results

We thank the reviewer for raising the point. The folate concentrations have been added to the table 1 of human studies.